# Recent Achievements in Medicinal and Supramolecular Chemistry of Betulinic Acid and Its Derivatives [note 2]

**DOI:** 10.3390/molecules24193546

**Published:** 2019-09-30

**Authors:** Uladzimir Bildziukevich, Zülal Özdemir, Zdeněk Wimmer

**Affiliations:** 1Institute of Experimental Botany of the Czech Academy of Sciences, Isotope Laboratory, Vídeňská 1083, 14220 Prague 4, Czech Republic; vmagius@gmail.com (U.B.); zulalozdemr@gmail.com (Z.Ö.); 2Department of Chemistry of Natural Compounds, University of Chemistry and Technology in Prague, Technická 5, 16628 Prague 6, Czech Republic

**Keywords:** betulinic acid, structural modification, supramolecular self-assembly, cytotoxicity, antitumor activity, antiviral activity, physicochemical parameters, ADME parameters

## Abstract

The subject of this review article refers to the recent achievements in the investigation of pharmacological activity and supramolecular characteristics of betulinic acid and its diverse derivatives, with special focus on their cytotoxic effect, antitumor activity, and antiviral effect, and mostly covers a period 2015–2018. Literature sources published earlier are referred to in required coherences or from historical points of view. Relationships between pharmacological activity and supramolecular characteristics are included if such investigation has been done in the original literature sources. A wide practical applicability of betulinic acid and its derivatives demonstrated in the literature sources is also included in this review article. Several literature sources also focused on in silico calculation of physicochemical and ADME parameters of the developed compounds, and on a comparison between the experimental and calculated data.

## 1. Introduction

Plants represent an important challenge in searching for new plant products, of which a majority of new structures displays drug-like properties in treating serious diseases. Medicinal plants represent a diverse and rich source of bioactive plant products [1]. In this review article, attention has been focused on betulinic acid and its derivatives. Betulinic acid is a very potent plant triterpenoid compound with a broad spectrum of its own activity and a broad spectrum of pharmacologically important derivatives. The most important disadvantage of betulinic acid is its very low solubility in aqueous media, which also indicates its low bioavailability. Regardless of this disadvantage, betulinic acid displays a spectrum of biological activity that includes cytotoxicity, antitumor activity, antiviral activity, anti-diabetic activity, anti-inflammatory activity, etc. [1,2]. More details on the mode of action of **1** can be found in the original literature cited here, and it is mentioned in each paragraph dealing with different types of activity. In the natural sources, betulinic acid—like all other triterpene acids—appears in forms of more polar conjugates formed mostly with mono- and oligosaccharides or sugar esters [3]. Targeted derivation of betulinic acid may result in designing compounds displaying more favored physicochemical and ADME parameters, enhancing the potential of practical applicability of the compounds in medicinal and supramolecular chemistry.

## 2. Plant Sources and Discovery of Betulinic Acid

Betulinic acid, (3β)-3-hydroxy-lup-20(29)-en-28-oic acid (**1**, Figure 1), was discovered in natural plant sources long ago. It was first isolated from *Gratiola officinalis* at the beginning of the 20th century under the trivial name graciolon [4]. At the very beginning, this plant product had been subsequently isolated from different plant sources, and, therefore, described by different trivial names (e.g., platanolic acid, cornolic acid, melaleucin, etc.), however, finally identified as betulinic acid (**1**, Figure 1) [5]. It is a naturally occurring pentacyclic lupane-type triterpene, found throughout the plant kingdom (e.g., in genera *Betulla*, *Ziziphus*, *Syzygium*, *Diospiros* or *Paeonia*) [6]. However, it has also been isolated from the bark of the plane tree (*Platanus acerifolia*) [5] or from the Western Australian Christmas tree (*Nuytsia floribunda*), where it is accompanied by a small amount of betulin (**2**, Figure 1), and from the barks of six *Melaleuca* species, *M. rhaphiophylla*, *M. cuticularis*, *M. viminea*, *M. leucadendron*, *M. parvijora*, and *M. pubescens*. The first five of the *Melaleuca* species belong to the group popularly known as ‘‘paper-barks”, and crystalline triterpenoid acid can be seen in places between the thin papery layers of the bark. Betulinic acid (**1**) has also been obtained from the inland form of dysentery bush (*Alyxia buxifolia*). Generally, **1** was found in different plants, both as a free aglycon and in forms of glycosylated derivatives that enhance its bioavailability [2,3]. Isolation of **1** from plant sources is not easy, because it is accompanied by a number of other terpene-based plant products. Betulin (**2**) is one of the most commonly occurring triterpene plant product often present in higher quantity than betulinic acid (**1**) itself. However, it can be converted into an aldehyde (**3**, Figure 1), followed by additional oxidation into **1** [7,8]. Purification of the target compounds can be achieved by a combination of different chromatographic methods, combined with crystallization, resulting in the white powdery compounds [5]. A new challenge may be seen in extracting the convenient plant material by supercritical carbon dioxide (SC-CO_2_) [9], both, without or with polarity modifier, or by pressurized liquid extraction (PLE) [10].

## 3. Pharmacological Effects of Betulinic Acid

### 3.1. Cytotoxicity and Antitumor Activity

Since 1970s papers appeared on the investigation of pharmacological effects of **1**. Cytotoxicity is one of the most important fields in this investigation, and it is one of the most widely investigated aspects of pharmacology of betulinic acid (**1**) [3]. At the very beginning, cytotoxicity of different plant extracts was observed, where **1** was later found in different mixtures of plant products [11,12]. Cytotoxicity of **1**, leading to apoptosis of tumor cells, was observed during the later studies [13,14,15,16]. Regardless of extensive investigation, the molecular target of **1** has not yet been identified. Speculations about the possible target(s) were based on pathway alterations like modulation of B-cell lymphoma (Bcl-2) and nuclear factor NF-κB, enhancer of activated B-cells, and antiangiogenic activity [17,18]. There is enough information about the activity of **1** and its potent derivatives, both in in vitro and in vivo models. However, it seems that in reality small changes in the chemical structure could lead to significant differences in specificity and mechanisms of action [19,20,21]. Nevertheless, the most recent studies indicate that the main mechanisms involved are the stimulation of apoptosis and the inhibition of kinases, both accompanied by a worthy antioxidant effect [1]. Betulinic acid (**1**) has been capable of reducing many of the toxicity indicators of the antitumor agent doxorubicin, which has been known to have strong cardiotoxic activity. These parameters, measured in human blood lymphocytes, include generation of reactive oxygen species, production of inflammatory cytokines, such as interleukin IL-12 (produced by B-lymphoblastoid cells) or tumor necrosis factor TNF-α (biosynthesized as prohormone with long and atypical signal sequence), alteration of mitochondrial membrane potential, and various morphological and histochemical changes to the apoptotic process [1].

However, it is important to stress that **1** used in these experiments consisted of a number of supramolecular aggregates formed by a process of self-assembly, which is typical for this type of triterpenoid substance in aqueous or hydroalcoholic media. These self-assembled particles were found to exert pro-apoptotic and genotoxic activity in K562 myelogenous leukemia cells, with higher efficacy than betulinic acid (**1**) itself [22].

### 3.2. Antiviral Activity

The absolute majority of original papers and recent reviews dealing with antiviral activity of **1** have been dealing with its anti-HIV activity [23,24]. This type of activity is a part of general antiviral activity, however, only a few reports appeared on the investigation of the general antiviral activity of **1** [5].

Human immunodeficiency virus (HIV)-caused HIV infection and acquired immunodeficiency syndrome (AIDS) were first identified over 30 years ago [25]. Global AIDS statistics estimate that 37 million people are living with HIV at present. Among them, 2 million people were newly infected with HIV, and more than 1 million died from AIDS-related illnesses [26]. Although over 30 drugs targeted at different steps of the viral life have been approved or are in experimental stages for treatment of HIV, remedy for HIV infection has not yet been found [27]. HIV therapy suffers from the rapid emergence of drug-resistant viral strains and harmful side effects caused by long-term drug treatment [27]. Therefore, a search for new and innovative anti-HIV agents has been an important research priority. Betulinic acid (**1**) represents a promising structure type for anti-HIV agents [26,28].

It acts against HIV by preventing the cleavage of the capsid-spacer peptide of the Gag protein, thereby impeding viral maturation. This causes the host cell releases virions with no infective capacity. The efficacy of **1** is influenced by various polymorphisms of this protein, especially in the residues 369–371 (QVT in wild type), with one of the best known as V370A [3]. Various structural derivatives of **1** were found to be more potent anti-HIV agents than betulinic acid (**1**) itself, however, those compounds will be discussed further in this review.

The effect of **1** against herpes viruses, especially against clinical strain HSV-1, is an example of other antiviral activity of **1** [29], the authors investigated and evaluated the in vitro experiments. They showed that after incubating the active principle with the virus, both sensitive and acyclovir-resistant strains lost their infectivity. In turn, neither pretreatment of the cell nor administrations at the time of viral propagation were effective [3,29].

Hepatitis B, which has an important health impact, represents another virus susceptible to betulinic acid (**1**). The inhibition of hepatitis B replication exerted by the triterpene is based on the downregulation of mitochondrial superoxide dismutase (SOD2) through the dephosphorylation (Ser133) of the cAMP response element-binding transcription factor at its binding site with the SOD2 promoter. Betulinic acid (**1**) has been shown to facilitate the translocation of the hepatitis B virus X protein into the mitochondria of mouse hepatocytes. The antiviral activity was strictly dependent on SOD2 because overexpression of this enzyme suppressed the effect [30].

### 3.3. Anti-Inflammatory Activity

The investigation of the anti-inflammatory activity of **1** was performed by focusing on the induction of inflammation by various activators of protein kinase C and other agents. Betulinic acid (**1**) inhibited the edema induced by toxic diterpene esters, mezerein, 12-deoxyphorbol-13-tetradecanoate, and 12-deoxyphorbol-13-phenylacetate, by 48, 51, and 61% (ID_50_ = 0.77 μmol/ear in this case), respectively, at a dose of 0.5 mg/ear [3]. Anti-inflammatory activity was also described in a macrolide lactone bryostatin-1-induced mouse ear edema (65% at 0.5 mg/ear), a peptide inflammatory mediator bradykinin-induced mouse paw edema (54% at 10 mg/kg), and rat skin inflammation induced by glucose oxidase (39% at 0.25 mg/site) [31,32]. In turn, no effect was observed in ear edema induced by arachidonic acid, resiniferatoxin, and xylene. Since **1** was inactive against arachidonic acid-induced inflammation, as well as in neurogenic inflammatory models, it is probable that this type of inflammation may depend on in vivo inhibition of protein kinase C [3,31,32].

### 3.4. Anti-Diabetic Activity

Anti-diabetic activity of betulinic acid (**1**) was described against type 2 of diabetes mellitus [33]. Based on different studies, both in vitro and in vivo, the mechanism of action of betulinic acid (**1**) was designed, reflecting glucose uptake, insulin resistance, insulin sensitivity, and glycogen biosynthesis. Those aspects were reviewed recently [3], and no substantial novel items of information have been found in the more recent literature data.

### 3.5. Anti-Hyperlipidemic Activity

When tested on obese mice, **1** was administered in drinking water for 15 days (50 mg·L^−1^). It caused decreasing of total triglycerides and cholesterol levels in high fat containing diet supplied to obese mice, which resulted in decreasing of mouse body weight, abdominal fat accumulation, blood glucose increase, and content of cholesterol and triglycerides in plasma [34]. Due to the action of **1,** an increase of insulin in plasma was observed [34]. Betulinic acid (**1**) was also able to reduce lipogenesis and lipid accumulation, which was observed during numbers of experiments in vitro and in vivo [35], in which the mechanism of action of **1** in anti-hyperlipidemic activity was studied and described.

### 3.6. Other Activity

The anti-parasitic and anti-infectious activity of **1** and its derivatives were also mentioned in the literature in recent time [3]. However, when searching for the details, we have found that these types of pharmacological effects were described either for various derivatives of betulinic acid (**1**) or for simultaneous treatments with betulinic acid (**1**) along with different compounds, mostly synthetic medicaments, acting in synergy. It is, therefore, no clear evidence that **1** itself is able to induce these types of activity [3]. These potential effects of **1** should be focused on more details in the future.

## 4. Supramolecular Characteristics of Betulinic Acid

It was already stated that **1** is the triterpene-type compound bearing a rigid pentacyclic lupane-type (6-6-6-6-5) backbone. Its molecule is 1.31 nm long [36]. Self-assembly of the molecules of **1** in aqueous media and organic solvents results in fibrous supramolecular systems. Those systems are able to present themselves as supramolecular gels. This behavior of **1** was tested with a number of organic solvents using different concentrations of **1** [36]. Since **1** is practically insoluble in water, gelling properties could not be proven in it, however, using ethanol/water mixtures (4:1 to 19:1) supramolecular gels were produced at a concentration *c* = 2.55% (*w/v*) [36]. Optical microscopy micrographs proved a formation of fibrous supramolecular systems in 1,2-dichlorobenzene [36]. More recent SEM studies of supramolecular self-assembly of **1** in mesitylene or chlorobenzene proved a formation of fibrous-like xerogels. Fibrous networks were also investigated by the AFM micrographs of **1** in *p*-xylene [37]. Self-assembly of **1** in different solvents was probably the first published example of the self-assembly of a triterpene molecule bearing no other substitution. Self-assembly characteristics of **1** in different liquids encouraged use of this type of supramolecular systems in various practical applications, namely selective damage of cancer cells without affecting the normal cells. This investigation of supramolecular behavior of **1** has been an important challenge in treating this fatally important disease [37,38,39].

## 5. Derivatives of Betulinic Acid, Their Pharmacological Effects, and Supramolecular Characteristics

### 5.1. Cytotoxicity and Supramolecular Characteristics

The area of monitoring cytotoxicity of different derivatives of **1** is well documented by 2015 by several review papers [1,15,39,40], as well as by a patent review [41]. Due to that fact, attention in this review article has been focused on the most recent period of several last years to cover novel betulinic acid derivatives not yet mentioned in the so-far published review papers, which have been found in the most recent literature sources. Older literature sources are cited when coherence in the text or in history requires such references.

Thus, a synthesis of novel 2,3-seco-triterpenoids and triterpenoids bearing five-membered ring A, all displaying cytotoxicity, was published (Scheme 1) [42]. The compounds were prepared on the basis of betulone (**4**) that is modified to the 2-oxime derivative of betulonic acid (**5**) by a several steps procedure [42,43]. Further structural modification of **5** (Scheme 1) resulted in a subsequent synthesis of the compounds **6**–**9** [42], representing the most cytotoxic structures of this series of compounds [42]. Their antiproliferative activity was tested on various human cancer cell lines. The compound **9** of this series showed adequate selectivity and cytotoxicity towards several important cancer cell lines, and, therefore, it was selected as the most active compound (IC_50_ = 3.4–10.4 µM for HEp-2, HCT116, RD TE32, and MS cancer cells). It was capable of triggering caspase-8-mediated apoptosis in HCT116 cancer cells accompanied by typical apoptotic chromatin condensation with no loss of mitochondrial membrane permeability.

To proceed with this investigation, a synthesis of a large series of novel hydrophilic esters of triterpenoid acids with cytotoxic activity was presented, and **1** was one of the natural products selected for the derivation [44]. Complex alcoholic groups (aliphatic or containing heterocycles), glycolic unit and monosaccharide groups were used as structural modifiers (Scheme 2). An easy synthetic procedure was described, in which the introduction of the above-identified substituents resulted in the preparation of the compounds with high cytotoxicity (**10a**–**10f**) [44]. Compound **10f** was the most selectively active compound of this series, effective against MCF7 (IC_50_ = 2.5 µM), HeLa (IC_50_ = 3.3 µM) and G-361 (IC_50_ = 3.4 µM) cancer cell lines.

Diamine and polyamine derivatives of betulinic acid were also investigated for their cytotoxicity and antimicrobial activity (Scheme 3) [45]. Derivation of **1** at the C(28)-carbon center is a several steps procedure (Scheme 3), leading through the intermediates **11**–**14c** to the target amides **15a**–**15c**. Derivation of **1** at the C(3)-OH group required protection and final deprotection of the C(28)-carboxyl group. Subsequent synthesis of the intermediates **16**–**19c** resulted in the preparation of the target structures **20a**–**20c** (Scheme 3). Their cytotoxicity, antimicrobial activity, and ability to form supramolecular self-assembled systems, preferably in aqueous media, were investigated and published [45,46]. The highest cytotoxicity values of these derivatives of betulinic acid were observed with the amides formed with piperazine and spermine. When the cytotoxicity of those compounds was studied, the amides **15a**–**15c** were found to display micromolar or even submicromolar activity (Table 1). However, they were toxic also towards the normal fibroblasts at a comparable level, and they show no or low selectivity. In turn, amides **20a**–**20c** showed selectivity and no toxicity towards the normal fibroblasts. Several of those new compounds showed also antimicrobial activity [45]. The investigation of diamines and polyamines as structural modifiers was then extended to steroids with the androstene skeleton [47], and important structure-activity relations have been found, resulting in observed selectivity of several prepared compounds [45,47].

Diamine- and polyamine-based amides of betulinic acid (**1**), i.e., the compounds **15a**–**15c** and **20a**–**20c**, were subjected to the detailed studies of supramolecular self-assembly [46]. The compounds **15c** and **20c** were found to form fibrous supramolecular structures, the formation of which was proven by the UV-VIS and DOSY-NMR measurements, and, subsequently, by AFM, SEM and TEM micrographs. It seems reasonable to say that a relation exists between cytotoxicity and supramolecular characteristics because the supramolecular structures and materials have a huge potential in the area of drug delivery [37,48]. The supramolecular structures formed from triterpenoid acid derivatives can act as a pharmacologically active delivery system to the target tissues where they can directly display their pharmacological activity.

The supramolecular characteristics of those compounds were discovered when the variable temperature pulsed-field gradient diffusion ordered NMR spectroscopy (VT-DOSY-NMR) was measured, in which the dependence of diffusion coefficient on temperature showed a non-linear curve [46]. DOSY-NMR spectroscopy is generally useful in determining the species of varying size formed by the studied compounds. This non-invasive pseudo-two-dimensional (2D) NMR technique has the potential to identify and virtually separate different aggregates existing simultaneously in the samples [49]. Application of this method to the self-assembled nanostructures has been undertaken during various investigations. However, the utilization of this technique for the supramolecular gels derived from low molecular weight gelators (LMWGs) remained still less explored [49,50,51]. In general, when the gel is formed, the diffusion coefficient should change in a non-linear way (i.e., non-linear dependence of diffusion coefficient on temperature appeared) compared to the situation in a clear solution (linear dependence was found) because of the decreased molecular mobility and increased viscosity of the gelled system (Einstein-Stokes equation) [52]. Both investigated compounds (**15c** and **20c**) displayed a non-linear dependence of the diffusion coefficient on the temperature of the measurement indicating a formation of supramolecular networks.

Subsequently, the compounds were subjected to a series of UV-VIS-NIR measurements, using a constant concentration of the studied compound in changing ratios of water/methanol mixtures in 10% steps, starting with pure water to pure methanol. Changes in the intensity of the peak maxima and, partly in wavelengths of those maxima also indicated a formation of supramolecular aggregates in the studied systems. The investigation was completed by visualizing the AFM, SEM, and TEM micrographs that proved a formation of fibrous supramolecular networks as well [46].

Picolyl amides of betulinic acid were investigated for their ability to cause tumor cell apoptosis (Scheme 4) [53]. The synthetic procedures described in the literature [53] in details, started at **1** and resulted in either of the target structures **22a**–**22c** or **24a**–**24c** (Scheme 4). While the picolyl amides of steryl hemiesters investigated in the past suffered from their low cytotoxicity [54], several of the picolyl amides of **1** showed high and even sub-micromolar effects [53]. The higher cytotoxicity was observed for the picolyl amine-based amides **22a**–**22c** in comparison with that of **24a**–**24c**. In the screening tests in G-361 human malignant melanoma cell line, **22b** showed therapeutic index TI = 100 (Table 2), which makes this derivative bearing the piperazine motif in the molecule to become one of the most important and most perspective compounds of this series [53].

Last two series of compounds derived from **1** [45,53] are the products of our team. We have also studied in silico calculations of physicochemical and ADME parameters of the target compounds **15a**–**15c**, **20a**–**20c**, **22a**–**22c**, and **24a**–**24c** (Appendix A). The basic details on the importance of these parameters are presented in the original papers [45,53], however, we would like to focus on the most important findings emerging from these calculated values. The recommended range for optimization of several physicochemical and ADME parameters appeared in the literature (cf. [45,53] and the literature cited therein). When comparing the recommended range for the selected parameters with their experimental data (Appendix A), the results can be summarized in this form: (a) molecular weight: only **15a** appears within the given range, (b) log *P*: all compounds appear out of the given range, (c) log *S*: **20b** and **24a**–**24c** are out of the given range, (d) the range for the number of H_acc_ and H_don_ is completed by all studied compounds, (e) all studied compounds meet the limits for log *PB* and log *BB*. The experimental results show that the range of molecular weight is less important, cytotoxicity was also found for compounds having MW > 500 (cf. Table 1 and Table 2). None of the studied compounds appeared within the given range for the parameter log *P*, it was not the driving parameter affecting the experimental cytotoxicity values. The parameter of solubility (log *S*) played more important role because neither **20b** nor **24a**–**24c** displayed high values of cytotoxicity. Compound **22c** appeared to be surprising exception for its very low cytotoxicity. The ADME parameters did not influence these result, because all studied compounds showed their calculated values of log *PB* and log *BB* within the given range. However, these ADME parameters are connected with central nervous system (CNS) active drugs, and none of the studied compounds appeared among potentially CNS active drugs [45,53].

Chinese authors [55] designed an interesting conjugate **29** derived from **1** with diazen-1-ium-1,2-dioxolate (**28**) and found an anticancer agent releasing nitric oxide (NO) and causing cancer cell apoptosis (Scheme 5). Antiproliferative activity of **29** was tested on several cancer cell lines even with sub-micromolar concentrations. The effect of the compound **29** was compared with that of cisplatin, and it was found that the synthesized compound **29** was 20 to 2 times more active than cisplatin against different cancer cell lines, while 3 times less active on reference liver cell line [55].

### 5.2. Antiviral Activity

#### 5.2.1. Anti-HIV Activity

Bevirimat (**30**, Scheme 6), a derivative of **1**, is a very potent maturation inhibitor of HIV-1. It interferes with the processing of P25 (CA-SP1) to CA, leading to the accumulation of P25 and producing immature HIV-1 particles. In 2007, bevirimat (**30**) succeeded in the phases I and IIa of clinical trials [56]. However, subsequent studies resulted in observation that its effectiveness was reduced in the treatment of 40−50% of patients who carried resistant viruses associated with naturally occurring polymorphisms in the SP1 region of HIV-1 Gag [57]. This finding resulted in blocking further clinical trials with **30** to treat HIV infection.

Modified bevirimat derivatives **31**−**34d** (Scheme 6) showed improved anti-HIV activity in comparison with **30** [58]. The procedures to synthesize **31**, **34a**−**34c**, and, subsequently, **34d** (Scheme 6) are described in the original literature [26]. The studied bevirimat analogs **31**−**34d** bear piperazine moiety in the molecule, which is quite often structural modification in drugs in general [59]. The so far made studies have resulted in a finding that piperazine can contribute to improving drug-like properties of the target compounds, namely bioavailability and metabolism [60,61]. Two nitrogen atoms located at the opposite positions in the piperazine molecule determine this compound as a convenient linker for merging desired structural motifs. Having different pharmacophores with different mechanisms of action in a single molecule may result in designing new compounds with enhanced efficacy [26]. Compound **34c** was found to be 3- to 50-times more potent than **30** against different types of the virus. A preliminary investigation of the mechanism of action indicated that **34c** is a HIV maturation inhibitor with good metabolic stability [26].

A series of C-3 phenyl- and heterocycle-substituted derivatives of the C-3 deoxybetulinic acid and C-3 deoxybetulin were also investigated for their potential as anti-HIV agents (Scheme 7) [62]. A 4-substituted benzoic acid moiety was identified as an advantageous replacement for the 3,3-dimethylsuccinate moiety present in **30**. The new analogs exhibit excellent in vitro antiviral activity against wild-type virus and a lower serum shift when compared with **30**. Compound **38** exhibits comparable cell culture potency toward wild-type virus as **30** (WT EC_50_ = 16 nM for **38** compared to 10 nM for **30**). However, the potency of **38** was less affected by the presence of human serum, while the compound displayed a similar pharmacokinetic profile in rats to **30**. Thus, **38** represents a new starting point for designing the second generation of HIV maturation inhibitors.

The effect of a spacer between the phenyl ring and the carboxylic acid was investigated by introducing linkers with different lengths and degrees of flexibility designed to study a wide range of structural motifs [62]. As already stated above, **38** displayed sub-micromolar inhibitory activity. The effect of replacement of the phenyl ring with a series of five- and six-membered heterocycles was investigated as an additional structural modification resulted in designing another successful subseries of active compounds shown in the formula **42** and the general formula **44**. These compounds (**42** and **44**) also displayed sub-micromolar activity values accompanied by high therapeutic indices [62].

Recently, a new class of α-keto amides of **1** was developed and identified as HIV-1 maturation inhibitors [63]. The compound **53** was identified with IC_50_ values of 17 nM (HIV wild type), 23 nM (Q369H), 25 nM (V370A), and 8 nM (T371A), respectively, as a leading structure of this series of compounds (Scheme 8). When tested in a panel of 62 HIV-1 isolates covering a diversity of CA-SP1 genotypes including A, AE, B, C, and G using a PBMC-based assay, **53** was potent against a majority of isolates demonstrating an improvement over the first generation maturation inhibitor, bevirimat (**30**) [63]. The data also demonstrated that **53** shows a mechanism of action consistent with inhibition of the proteolytic cleavage of CA-SP1 [63].

Amides of the general formulae **54**−**57** (Figure 2), incorporating a basic side chain, provided excellent potency against both wild type and V370A viruses while maintaining a low human serum shift [64,65]. In addition, **54a** exhibited an EC_50_ = 31 nM against the ΔV370 Gag polymorphism. In turn, the structures **54** exhibited low oral exposure, attributed to a combination of poor solubility and low membrane permeability, precluding their further advancement [65]. To improve the disadvantages of the structure **54**, the structures **55** and **56** were developed. However, their antiviral activity did not increase in comparison with those of **54**. To proceed with improving the disadvantages of **54**−**56** again, the dibasic C-28 amine **57a** and monobasic **57b** were designed. Later on, an optimized structure **57c** showed improved antiviral profile in screening tests against three viruses [65]. However, **57c** became a subject of additional structural modification resulting finally in **57d**, which demonstrated targeted antiviral activity and improved oral exposure in rats. Installing the more basic amine closer to the lipophilic core had the effect of shielding the NH and may have a positive impact on the permeability properties of the molecule and, ultimately, on oral exposure [64,65].

A concise and scalable second-generation synthesis of HIV maturation inhibitor **64** (Scheme 9) was published in 2017 [66]. The synthesis was based on an oxidation strategy involving a CuI mediated aerobic oxidation of betulin (**2**), a highly selective PIFA mediated dehydrogenation of an oxime, and a subsequent Lossen rearrangement, which occurred through a unique reaction mechanism for the installation of the C17 amino functionality. The synthetic procedure consisted of seven steps with a 47% overall yield and it begins from the abundant and inexpensive natural product betulin (**2**) (Scheme 9) [66]. The target compound **64** became a potent HIV-1 inhibitor in cell culture that exhibited a broad spectrum of antiviral effects that encompass the V370A- and ΔV370-containing polymorphic viruses. In addition, **64** exhibited low serum binding, which resulted in the modest effect on potency in vitro, and in a preclinical suggestion of dosing once daily in humans. In the phase IIa of clinical trial, 10-days of monotherapy with two administered doses daily to the treatment-non-experienced subjects and treatment-experienced subjects, both infected with HIV-1 subtypes B or C, was generally safe and well-tolerated, and it demonstrated important reduction in viral RNA [65]. Compound **64** has currently been evaluated in the phase IIb of clinical study as a part of a treatment regimen with mechanistically different antiretroviral agents. The so far achieved values of practical importance with **64** in the HIV-1 treatment are WT EC_50_ = 1.9 nM (HIV-1 WT), and EC_50_ = 10.2 nM (HIV1 WT (HS)), which are very promising values.

#### 5.2.2. Antiherpetic Activity

Herpes simplex virus types 1 and 2 (HSV-1 and HSV-2) represents other types of virus that may be treated by derivatives of betulinic acid. The developed ionic derivatives represent betulinic acid structural modification capable of improving water solubility and biological activity of the target structures **65a**–**66b** (Figure 3) [67]. The binding properties of these derivatives with respect to the human serum albumin (HSA) was examined and found to be similar to current anti-HIV drugs. These compounds (**65a**, **65b**, **66a**, and **66b**) inhibited HSV-2 replication at concentrations similar to those reported for acyclovir (IC_50_ 0.1–10 µM) and with minimal cellular cytotoxicity. IC_50_ values for antiviral activity against HSV-2 186 were 1.6, 0.9, 0.6, and 7.2 µM for the compounds **65a**–**66b**, respectively. However, these compounds did not inhibit HIV reverse transcriptase. Compound **66a** was the most active compound of this series in treating HSV-2.

Amide conjugates with four structural types of β-amino alcohols were synthesized from 2,3-seco-18α*H*-oleananoic and 2,3-seco-lupane C-3(C-28) mono- and dicarboxylic acids, in order to prepare novel agent for treating herpes simplex virus, types 1 and 2 (Scheme 10) [68]. Esters were prepared by a reaction of C(3)-hydroxy derivatives of A-seco-triterpenoids with dicarboxylic acid anhydrides. The antiviral activity of the synthesized compounds was studied against HSV-1. The most active amide (**68**) displayed antiviral effect EC_50_ = 5.7 µM). In contrast, the effect of **67** was about one order of magnitude lower than that of **68**.

To continue a search for agents active against herpes simplex virus, unsaturated acids, including difficultly accessible ceanothane-type ones, were prepared by the same authors, using alkaline hydrolysis of semi-synthetic triterpenoids with 1-cyano- or 3-methyl-1-cyanoalkene fragments in five-membered ring A (Scheme 11) [69]. Regioselective reduction of 1-cyano-19β,28-epoxy-2-nor-18α*H*-olean-1(3)-ene by DIBAL-H resulted in a synthesis of 2-aminomethyl-19β,28-epoxy-2-nor-18α*H*-olean-1(3)-ene. The synthesized triterpene derivatives possessed antiviral activity against HSV-1. Compound **70** (ET_50_ = 17.7 µM) was the most active compound of this small series, while **71** was less active.

#### 5.2.3. Antihepatitic Activity

Hepatitis represents another type of virus affecting the human population. It has often been treated with oleanane derivatives (e.g., **73**, Scheme 12) [70], but betulinic acid has also been used for treating this disease [71]. Epstein-Barr virus (EBV), responsible for hepatitis and a number of other similar diseases, has widely infected more than 90% of human populations. Currently, there is no efficient way to remove the virus because the EBV carriers are usually in a latent stage that allows them to escape the immune system and common antiviral drugs. In the effort to develop an efficient strategy for the removal of the EBV virus, **1** has been shown to suppress EBV replication through SOD2 suppression slightly, with subsequent reactive oxygen species generation and DNA damage in EBV-transformed lymphoblastoid cell lines. Chidamide is a novel synthetic histone deacetylase inhibitor capable of switching EBV significantly from its latent stage to the lytic stage with increased gene expression of BZLF1 and BMRF1. However, it has a small effect on EBV replication due to the suppression effect of chidamide-mediated reactive oxygen species generation. Interestingly, a combination of **1** and chidamide synergistically inhibits EBV replication with ROS over-generation and subsequent DNA damage and apoptosis [72]. Overexpression of SOD2 diminishes this effect, while SOD2 knockdown mimics this effect. An in vivo tumor development study with the tail vein injection of EBV-transformed lymphoblastoid cell lines in nude mice proved that the combination of **1** with chidamide synergistically increased superoxide anion release in tumor tissues and suppressed EBV replication and tumor growth, prolonging significantly mouse survival. The combination of **1** with chidamide (Figure 4) could be an efficient strategy for clinical EBV removal. Unfortunately, the Chinese authors [72] did not mention clearly if they tested a mixture of **1** with chidamide that can possibly result in a formation of a pyridinium salt **74** or if they prepared the amide **75** and tested that chemical species. At the moment of finalizing this manuscript, no more details were accessible in the literature.

## 6. Conclusions

Betulinic acid (**1**) and its broad spectrum of derivatives demonstrated clearly their power in different areas of human diseases that can be successfully treated with those compounds based on a single plant product. In all areas of pharmacological applications of **1**, further development and challenge for designing novel structures are still needed. Regardless of the number of successfully active derivatives of **1**, there are still rare structures that have found their practical application in medicine. We believe that a synergic investigation of pharmacological effects and supramolecular characteristics may lead to a better understanding of the mechanism of action of those natural and semisynthetic compounds in at least several areas of treating fatal diseases mentioned in this text.

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
