# Peer review of "Recent Achievements in Medicinal and Supramolecular Chemistry of Betulinic Acid and Its Derivatives [Author-notes fn2-molecules-24-03546]"

_molecules, 2019, doi:10.3390/molecules24193546_

Round 1

Reviewer 1 Report

This is a well-written and very comprehensive review of the synthesis and described biological activity of the derivatives of betulinic acid, a triterpenoid compound originally isolated from a number of tree species. The authors give a short review of its isolation and characterization and as well as a short review of the natural products biological activity covering their cytotoxicity, antiviral, anti-inflammatory, anti-diabetic, and anti-hyperlipidemic activities. The bulk of the review focuses on the characterization of the synthetic derivatives of betulinic acid, in which is found more enhanced biological activates and the characterization of its supramolecular forms in solution. I found this review to be extremely comprehensive and up to date. The focus is on the more recent research work of the past three years, but the authors cite a number of key reviews to cover the decades of work that have preceded this review. Much of the synthetic work and biological e=activity assessment is outside my area of expertise, so I did not see any major or minor area that were not addressed nor anything that needed to be added. I consider this to be an important addition to the literature and should be accepted for publication.

Author Response

Reviewer No. 1:

Thank you for your valuable evaluation of our manuscript. No comments appeared in your review needing answers.

Reviewer 2 Report

In their manuscript Bildziuchevich et al. present an overview of the recent achievements reached on the pharmacological activity and supramolecular features of betulinic acid and its derivatives. In particular authors discuss the anti-tumor, anti-diabetic and anti-viral properties of these family of natural compounds. The review is well written, but some points need more explanations.

The description of cytotoxic effect, since it is referred to cancer cell systems, should be reported as antitumor action in the abstract and introduction section. In particular, authors should report in details the known molecular mechanisms and proteins involved in the description of the biological activity of betulinic acid and its derivatives. A synthetic table summarizing the multifaceted actions of betulinic acid should be included reporting the experimental models on which betulinic acid has been assayed, the molecular targets and observed effects, as well as the corresponding literature sources. Because of the low solubility of betulinic acid, its application in vivo systems could be strongly limited. This aspect, as well as other possible alternatives to guarantee its delivery to target tissues, should be discussed by authors.

Author Response

Reviewer No. 2:

Thank you for your valuable evaluation of our manuscript. The term antitumor activity was added to the abstract, among the key words and to the Introduction. Furthermore, a general reference was added into the Introduction, referring to the literature sources in each chapter in this manuscript. The original literature cited therein gives a more detailed information on the modes of action of betulinic acid and on its derivatives, if known in details. It is clearly stated in the manuscript that structural modification of betulinic acid often results in changes in mode of action of the target derivative(s). Solubility and bioavailability has often been a target of synthetic chemists to modify the properties of betulinic acid derivatives in comparison with those of betulinic acid in a way to improve those characteristics and, therefore, improve bioavailability of the target derivative(s). It has been discussed in different paragraphs of the manuscript.

Reviewer 3 Report

This is an excellent up to date about betulinic acid derivatives and their pharmacological properties.

In my opinion shoul be published as it is, aftyer two very minor additions.

Please include the numbering for betulinic acid in order to understand better some reactivity.

Please include this reference: Rene Csuk, Expert Opin. Ther. Patents (2014) 24(8)

Author Response

Reviewer No. 3:

Thank you for your valuable evaluation of our manuscript. We have included carbon atom numbering in the structure of betulinic acid in Figure 1. The recommended reference was included into the manuscript under the current number [41]. Unfortunately, we do not have full access to the article of Prof. Csuk [41], only its abstract was available in our institutional library on-line. All literature sources following the new number [41] were re-numbered starting from [42].